# Intracellular accumulation of free cholesterol in macrophages triggers a PARP1 response to DNA damage and PARP1 impairs lipopolysaccharide-induced inflammatory response

**Kenneth K.Y. Ting[1,2], Hisham M. Ibrahim[2,3], Nitya Gulati[2], Yufeng Wang[2], Jonathan V. Rocheleau[2], Myron I. Cybulsky[1,2,3,4‡] ***

**1** Department of Immunology, University of Toronto, Toronto, Ontario, Canada, **2** Toronto General Hospital Research Institute, University Health Network, Toronto, Ontario, Canada, **3** Department of Laboratory Medicine and Pathobiology, University of Toronto, Toronto, Ontario, Canada, **4** Peter Munk Cardiac Centre, University Health Network, Toronto, Ontario, Canada

‡ Lead Contact
* myron.cybulssky@utoronto.ca

## Abstract

The formation of macrophage (Mφ) foam cells is a hallmark of atherosclerosis, yet how the process of lipid loading can modulate Mφ inflammatory responses by rewiring their intracellular metabolic circuits is not well understood. Our previous studies have shown that the accumulation of oxidized LDL (oxLDL) or free cholesterol in Mφs impaired their inflammatory response by suppressing HIF-1α-mediated glycolysis and upregulating NRF2 antioxidative response. However, it remains unclear if other metabolic processes are also contributory. In this study, we found that the accumulation of free cholesterol, but not oxLDL, in primary murine thioglycolate-elicited peritoneal Mφs (PMφs) enhanced a PARP1-dependent response associated with repair of DNA damage, which was characterized by poly ADP-ribosylation of proteins, phosphorylation of histone 2A.X and consumption of $NAD^+$. Both oxLDL and cholesterol enhanced the PARP1 response after LPS stimulation. Treatment of PMφs with mitoTEMPO, a specific mitochondrial reactive oxygen species (mtROS) scavenger, alleviated mtROS during cholesterol loading, blocked the PARP1 response and partially restored LPS- induced inflammatory gene expression. In contrast to inhibition of PARP1 enzymatic activity, knockdown of PARP1 expression in RAW264.7 Mφs with siRNA elevated LPS-induced inflammatory gene expression. Overall, our study suggests that cholesterol accumulation triggers a PARP1 response to DNA damage in Mφs and that PARP1 inhibits LPS-mediated inflammation through a non-enzymatic function.

## Introduction

Atherosclerosis, a chronic inflammatory disease characterized by the recruitment of inflammatory monocytes to the intima, remains one of the leading causes of vascular disease

**Data availability statement:** All relevant data are within the manuscript and its Supporting Information files.

**Funding:** This study was supported primarily by the Canadian Institutes of Health Research (CIHR) grant FDN-154299 (M.I.C.). M.I.C. holds a Tier 1 Canada Research Chair. K.K.Y.T was supported by fellowships from Canadian Institutes of Health Research, Ontario Graduate Scholarship, fellowships from the University of Toronto and The Peterborough K. M. Hunter Charitable Foundation. The funders had no role in study design, data collection and analysis, decision to publish, or preparation of the manuscript.

**Competing interests:** The authors have declared that no competing interests exist.

worldwide. The recent success of the CANTOS trial, which targets the inflammatory interleukin-1β pathway with canakinumab, demonstrated therapeutic benefit in patients with advanced atherosclerosis independent of lipid-level lowering [1]. The result from this study thus supports the notion that in addition to hypercholesterolemia, inflammation itself also plays a critical role in driving the pathogenesis of atherosclerosis. Therefore, understanding the cellular processes that orchestrate the inflammatory process of atherosclerosis is crucial for the development of effective therapeutics.

Hypercholesterolemia is a key risk factor of atherosclerosis and the formation of foam cells, specifically, the appearance of lipid-laden intimal macrophages (Mφs), marks the beginning of atherogenesis [2]. While lipid loading of intimal Mφs is critical for the development of nascent atherosclerotic lesions [3], it remains unclear how this process is linked to the initiation of inflammatory events in atherosclerosis, such as the recruitment of monocytes. The traditional paradigm in the field suggests that lipid loading of Mφs is intrinsically an inflammatory process [4,5], although recent *in vivo* transcriptomic analysis of murine and human atherosclerotic lesions have revealed that foamy Mφs were less inflammatory than non-foamy Mφs [6,7]. These findings implicate that lipid loading of Mφs is insufficient to trigger inflammation and suggest the notion that other stimuli, potentially from the endothelium, are required to initiate inflammation in atherosclerosis [8].

To model the formation of foam cells in atherosclerotic lesions *in vitro*, we have incubated primary murine thioglycolate-elicited peritoneal Mφs (PMφs) with oxidized low-density lipoproteins (oxLDL) or free cholesterol, an established and reproducible method used to induce the formation of lipid droplets in Mφs [8,9]. Utilizing this model, our group and others have consistently reported that lipid loading of Mφs with either oxLDL or cholesterol alone is insufficient to drive inflammation in Mφs [8–10]. More importantly, we found that the inflammatory responses of lipid-loaded Mφs were in fact suppressed upon their activation by an inflammatory stimulus, such as LPS [9–12]. Mechanistically, the dampening of the inflammatory response was dependent on multiple mechanisms, including the impairment of p65 binding to the promoters of pro-inflammatory genes [9], the suppression of HIF-1α-dependent activation of inflammatory genes, as well as the overactivation of NRF2-dependent antioxidative genes [10]. Understanding that the inflammatory response of Mφs is also dependent on the rewiring of their metabolism, we have previously characterized the metabolic profiles of oxLDL and cholesterol-loaded Mφs and found that their glucose metabolism (glycolysis) was impaired in a HIF-1α-dependent manner, while NADPH consumption was repurposed for anti-inflammatory pathways in a NRF2-dependent manner [10,13,14]. It is possible that other metabolic processes may contribute to dampening the inflammatory response in lipid-loaded Mφs.

Poly [ADP-ribose] polymerase 1 (PARP1), the first member of the diphtheria toxin-like ADP-ribosyltransferase (ARTD) family of enzymes to be identified, plays an important role in initiating the repair of single-strand DNA breaks by various stress sources, including reactive oxygen species (ROS) [15]. In general, PARP1 is responsible for catalyzing the addition of linear or branched poly-ADP-ribose (PAR) polymers to itself or other target proteins by consuming NAD$^+$ [15]. In the context of DNA damage repair, PARP1 is rapidly recruited to the sites of DNA damage, which are enriched with the phosphorylation of H2A histone family member X (γ-H2A.X), and subsequently PARP1 catalyzes the formation of PAR on itself, as well as histone proteins [15]. The formation of PAR polymers helps to recruit proteins that are involved in repairing DNA damage and DNA metabolism as they can interact with PAR polymers covalently and non-covalently [15]. Aside from its established role in maintain genome integrity, PARP1 has also been shown to perform alternative functions, one of which is serving as a co-activator for transcription factors, such as NF-κB

[16] and NRF2 [17], thereby demonstrating that PARP1 can be a regulator of inflammatory responses. More recently, the inflammatory role of PARP1 has been revealed in activated Mφs as NLRP3 inflammasome-mediated IL-1β production was inhibited in LPS-stimulated Mφs with *Parp1* genetic deficiency [18]. Similarly, the inflammatory responses of Mφs activated with *T. cruzi*-derived extracellular vesicles were also impaired upon chemical inhibition or genetic deletion of *Parp1* [19]. Taken together, these results implicate the possibility that PARP1 can be a critical dichotomous regulator of Mφ anti-inflammatory and inflammatory responses.

In this study, we have characterized the metabolic profiles of PMφs loaded with oxLDL or cholesterol and found that loading with lipid significantly enhanced PARP1-mediated DNA damage repair response following LPS stimulation. This is associated with an increase synthesis of nucleotides through the salvage pathways, increased consumption of $NAD^+$, as well as induction of γ-H2A.X. Mechanistically, alleviating mitochondrial-derived ROS (mtROS) in cholesterol-loaded PMφs with mitoTEMPO, a mtROS specific scavenger, reduced PARP1 activation and restored inflammatory responses. Knockdown of PARP1 expression in RAW264.7 Mφs with siRNA elevated LPS-induced inflammatory gene expression in cells without cholesterol accumulation.

## Materials and methods

### Mouse strains

Male and female mice C57BL/6J (Strain #000664, Jackson Labs) 8 – 12 weeks-old were used. Mice were maintained in a pathogen-free, temperature-regulated environment with a 12-hour light and dark cycle and fed a normal chow diet (NCD, 16 kcal% fat). Mice used in comparative studies were gender and age matched. Studies were performed under the approval of Animal User Protocols by the Animal Care Committee at the University Health Network according to the guidelines of the Canadian Council on Animal Care. Specifically, mice were sacrificed with carbon dioxide under isoflurane-based anesthesia. The chamber is gradually filled with carbon dioxide to 40% of volume as an effort to alleviate suffering. Secondary physical method was used to confirm death.

### Thioglycolate-elicited peritoneal Mφ (PMφ) isolation

Mice were injected intraperitoneally with 1 mL of 4% aged thioglycolate (ThermoFisher Cat#211716) and PMφs were harvested after 4 days by lavage with cold PBS containing 2% FBS. Cells were counted and cultured (37°C, 5% $CO_2$) in DMEM supplemented with 10% FBS, 2 mM L-glutamine, 10,000 U/mL penicillin/streptomycin). Adherent PMφs were used in experiments after 18 h.

### Transient transfection of RAW264.7 Mφs

RAW264.7 Mφs (2 x 106, ATCC, Cat# ATCC TIB-71) were electroporated (Amaxa® Cell Line Nucleofector® Kit V, LONZA; Cat#VCA-1003) with 0.5 µM of scrambled (ThermoFisher, Cat#:4390843) and mouse PARP1 siRNA (Dharmacon, Cat#:L-040023-00-0005). Transfected cells were seeded in 6-well plates and cultured in recovery medium (DMEM, 20% FBS) for 3 h, then DMEM, 10% FBS for 48 h.

### Lipid loading, LPS stimulation and inhibitor studies

PMφs were cultured for 24 h with human medium oxidized low-density lipoprotein (100 µg/mL, Kalen Biomedical Cat#770202) or cholesterol (50 µg/mL, Sigma Cat#C3045),

followed by ultrapure LPS stimulation (10 ng/mL, InvivoGen, Cat#tlrl-3pelps) for up to 6 h. Ethanol (0.5%) was used as a carrier control for cholesterol. For inhibitor experiments, AG-14361 (100 nM) (Selleckchem, S2178) was added 1 h prior to LPS stimulation. Diphenyleneiodonium (chloride) (500 nM) (Cayman Chemical, Cat#81050), MitoTEMPO Hydrate (50 µM) (Cayman Chemical, Cat#16621) was added 1 h prior to oxLDL or cholesterol loading. These inhibitors were removed by washing prior to LPS stimulation.

## Immunoblotting

PMφs were cultured in each well of 12-well plates at $2 \times 10^6$ per experimental condition. Cells were lysed in ice-cold RIPA buffer (1% NP40, 0.1% SDS, 0.5% deoxycholate in PBS, supplemented with 1 mM PMSF, 1X cOmplete™, EDTA-free Mini Protease Inhibitor Cocktail (Sigma Cat#11873580001) and 1X PhosSTOP™ (Sigma Cat#4906845001)) for 15 minutes. Protein concentrations in lysates were determined by Protein Assay Dye Reagent (BioRad Cat#5000006), diluted in 2x Laemmli sample buffer (BioRad Cat#161-0737) with fresh β-mercaptoethanol (BioRad Cat#1610710), and heated at 95°C for 5 minutes. Samples (20 µg of protein per lane) were resolved on 8%-15% SDS-PAGE gels and transferred to polyvinylidene difluoride membranes (Sigma Cat#IPVH00010) using a wet transfer system. Membranes were blocked with 5% skim milk non-fat powder or 3% BSA (Bioshop Cat#ALB003) in Tris-buffered saline-Tween (TBST) for 1 h at room temperature. Membranes were incubated with primary antibodies overnight: anti-Actin (Sigma, A2066), anti-p53 (CST#32532), anti-Lamin A/C (CST#2032), anti-Mono/Poly-ADP Ribose (CST#83732), anti-PARP1 (CST#9542), anti-pH2A.X (Ser139) (CST#2577), anti-H2A.X (CST#2595) and anti-ABCA1 (PA1-16789, ThermoFisher), followed by washing and incubation with HRP-conjugated anti-rabbit IgG (CST#7074) (22°C, 1 h). Blots were developed using Immobilon Forte Western HRP substrate (Sigma, WBLUF0100), imaged with Microchemi 4.2 (BioRad) and analyzed with ImageJ. Original uncropped blots are provided (see Supporting Information).

## Extracellular oxygen consumption rate (OCR) measurement

PMφs ($2.5 \times 10^5$) were cultured in XF24 well plates (Agilent Technologies, Cat#102342-100), incubated with oxLDL for 24 h, followed by the injection of LPS by the Seahorse analyzer. Real-time OCR values were acquired and normalized (baselined OCR %) to one reading before LPS injection (as indicated by arrow).

## Quantification of NAD⁺ levels

PMφs ($2 \times 10^6$) and RAW264.7 Mφs ($3 \times 10^5$) were plated in each well of 12-well plates. PMφs were loaded with oxLDL or cholesterol for 24 h, followed by LPS stimulation for 3 h, then were washed with kit-provided assay buffer (Cayman, 600480) and lysed with 350 µl of the kit-provided permeabilization buffer (Cayman, 600480). Quantification of NAD⁺ levels were then determined by following manufacturer's protocol. Specifically, the reaction mix was incubated with lysates mix for 30 minutes at room temperature before quantification of absorbance at 450 nm with spectrophotometer. The consumption rate of NAD⁺ was measured by adding FK-866 (100 µM) (Cayman Chemical, Cat#13287) to cells every 15 minutes for up to 1 h. The consumption rate of NAD⁺ by PARP1 was measured by treating cells with FK-866 (100 µM) and AG-14361 (1 µM) or rucaparib (20 µM) to cells every 15 minutes for up to 1 h.

## Quantification of glucose-6-phosphate dehydrogenase activity

PMφs ($2 \times 10^6$) plated in each well of a 12-well plate were loaded with oxLDL for 24 h, followed by LPS stimulation for 6 h, then were lysed with 1% Triton X-100 in PBS. The activity of G6PD in lysates was quantified by following manufacturer's protocol (Cayman, 700300).

## RNA isolation and real-time (RT) PCR

Total RNA was isolated with E.Z.N.A.® Total RNA Kit I (Omega Cat#R6834-01) and reverse transcription (RT) reactions were performed with High-Capacity cDNA Reverse Transcription Kit (ThermoFisher Cat#4368814) according to manufacturer's protocol. Real time quantitative PCR (qPCR) was then performed using a Roche LightCycler 480 with Luna® Universal qPCR Master Mix (New England Biolabs, Cat#M3003E) on a white plate with adhesive sealing. Quantification of mRNA was performed by using primers that span over two adjacent exons, quantified using the comparative standard curve method and normalized to hypoxanthine phosphoribosyltransferase (HPRT), a housekeeping gene. Specifically, for each qPCR reaction, 5 μl of the Luna Universal qPCR Master Mix, 4 μl of < 100 ng of cDNA template, 0.25 μl of each forward primer (10 μM) and reverse primer (10 μM) is added. The sequence of qPCR primers and other relevant information are listed below:

| Gene (Gene ID) | Forward Primer | Reverse Primer | Exon location |
|---|---|---|---|
| *Il1a* (16175) | ACGGCTGAGTTTCAGTGAGACC | CACTCTGGTAGGTGTAAGGTGC | Exon 4–5 |
| *Il1b* (16176) | AGTTGACGGACCCCAAAAGA | TGCTGCTGCGAGATTTGAAG | Exon 3–4 |
| *Il6* (16193) | CTCCCAACAGACCTGTCTATACCA | TGCCATTGCACAACTCTTTTCT | Exon 2–3 |
| *Il12b* (16160) | AAGTGGGCATGTGTTCC | TCTTCCTTAATGTCTTCCACTT | Exon 7–8 |
| *Ccl3* (20302) | CCCAGCCAGGTGTCATTT | AGTTCCAGGTCAGTGATGTATTC | Exon 2–3 |
| *Ccl5* (20304) | CCTGCTGCTTTGCCTACCTCTC | ACACACTTGGCGGTTCCTTCGA | Exon 2–3 |
| *Ccl9* (20308) | TCCAGAGCAGTCTGAAGGCACA | CCGTGAGTTATAGGACAGGCAG | Exon 2–3 |
| *Ccl22* (20299) | GTGGAAGACAGTATCTGCTGCC | AGGCTTGCGGCAGGATTTTGAG | Exon 2–3 |
| *Tnfa* (21926) | GTAGCCCACGTCGTAGCAAAC | GCACCACTAGTTGGTTGTCTTTGA | Exon 3–4 |
| *Nos2* (18126) | AAACCCCTTGTGCTGTTCTC | GGGATTCTGGAACATTCTGTGC | Exon 2–3 |
| *Gclc* (14629) | ACACCTGGATGATGCCAACGAG | CCTCCATTGGTCGGAACTCTAC | Exon 10–11 |
| *Gclm* (14630) | TCCTGCTGTGTGATGCCACCAG | GCTTCCTGGAAACTTGCCTCAG | Exon 6–7 |
| *Gsr* (14782) | GTTTACCGCTCCACACATCCTG | GCTGAAAGAAGCCATCACTGGTG | Exon 5–6 |
| *Hmox1* (15368) | CACTCTGGAGATGACACCTGAG | GTGTTCCTCTGTCAGCATCACC | Exon 1–2 |
| *Nqo1* (18104) | GCCGAACACAAGAAGCTGGAAG | GGCAAATCCTGCTACGAGCACT | Exon 3–4 |
| *Txnrd1* (50493) | AGTCACATCGGCTCGCTGAACT | GATGAGGAACCGCTCTGCTGAA | Exon 6–7 |
| *Parp1* (11545) | CTCTCCCAGAACAAGGACGAAG | CCGCTTTCACTTCCTCCATCTTC | Exon 9–10 |
| *Hprt* (15452) | CTGGTGAAAAGGACCTCTCGAAG | CCAGTTTCACTAATGACACAAACG | Exon 7–8 |

The experimental details (step, temperature, time and cycles) for qPCR reactions are listed below:

| Cycle step | Temperature | Time | Cycles |
|---|---|---|---|
| Initial denaturation | 95°C | 60 seconds | 1 |
| Denaturation | 95°C | 15 seconds | 45 |
| Extension | 60°C | 30 seconds ( + plate read) | |
| Melt Curve | 60–95°C | Various | 1 |

## Immunofluorescence microscopy

PMφs ($3 \times 10^6$) were first seeded in 35 mm petri dish, with 14 mm microwell (MatTek, P35G-1.5-14-C), then cultured with cholesterol or oxLDL overnight. Cells were stimulated with LPS

for 3 h the next day, washed three times with pre-warmed HBSS (Wisent, Cat#311-513-CL). To detect γ-H2A.X, cells were fixed with 4% PFA for 1 h at 4°C, permeabilized with 0.5% Triton-X in PBS for 10 minutes at RT and blocked with 10% donkey-serum in PBS for 1 h at RT. Samples were incubated with primary monoclonal antibody against γ-H2A.X (ThermoFisher, Cat# MA5-27753) overnight. Next day, excess primary antibodies were washed off with PBS (3 washes, 5 minutes each) followed by incubation of donkey anti-mouse Cy3 secondary antibody (Jackson Immunoresearch, Cat# 715-167-003) for 1 h at RT. Excess secondary antibodies were washed off with PBS (3 washes, 5 minutes each) followed by addition of 32.4 μM of Hoechst nuclear staining reagent (ThermoFisher #H3570) to label nuclei. Cells were imaged with an inverted Nikon A1R laser scanning confocal microscope. Images were acquired using a 60x oil objective (N.A. 1.42) in Galvano mode (1024x1024 pixels). Quantification of γH2A.X was performed using Imaris 10.1 (Oxford Instruments, Abingdon, UK) using the "spots" tool to identify different spot sizes of γH2A.X foci. Quality and intensity mean filters were used to identify foci and exclude nonspecific positive pixels. The number and average area of foci was obtained from the statistics tab within the imaging software.

For ROS assay, cells were cultured for 1 h at 37°C, 5% $CO_2$ in HBSS, supplemented with mitoSOX (5 μM, ThermoFisher #M36008) and Hoechst 33342 nuclear staining reagent. Cells were then washed three times with pre-warmed HBSS and fixed with 4% PFA for 1 h at 4°C prior to imaging. Cells were imaged with A1R Confocal microscope with resonant scanner (Nikon). Mean fluorescence intensity measurements represented the ratio of total fluorescence intensity for each field to the number of nuclei in that field.

## Statistical analysis

The statistical test(s) used in each experiment is listed in the figure legends. In general, the figures show pooled data from independent experiments. All experiments were repeated at least three times, and the number of biological replicates is indicated as the n value. Statistical analyses were performed using the Prism software, unless otherwise specified in the figure legends.

## Results

### Intracellular accumulation of oxLDL induces DNA damage repair responses and modulates nucleotide metabolism in LPS-stimulated PMφs

Late-stage (> 4h) glycolytic reprogramming in LPS-stimulated Mφs is critically dependent on HIF-1α-mediated transcriptional regulation of glycolytic and inflammatory genes, such as *Il1b* [20]. Although our past study has shown that both cholesterol and oxLDL loading of PMφs suppressed LPS-induced glycolysis [10], it remains unclear if other metabolic processes are also modulated. To investigate this, we re-analyzed the transcriptome of oxLDL-loaded PMφs 6 h after LPS stimulation and found that oxLDL loading differentially modified metabolic and inflammatory processes compared to unloaded controls (Fig 1A). In agreement with our past findings, hallmark pathway analysis confirmed that oxLDL loading impaired LPS-induced inflammatory responses (i.e., TNFA_SIGNALING_VIA_NFKB_signal, IL6_JAK_STAT3_SIGNALING_signal), thus supporting the notion that oxLDL loading of PMφs suppresses Mφ inflammatory processes. On the other hand, oxLDL loading upregulated oxidative phosphorylation and DNA repair responses. Notably, certain genes in the DNA_REPAIR_signal pathway, such as *Dut* and *Umps*, were upregulated by oxLDL, suggesting that nucleotide metabolism is modulated to support an increased demand for nucleotides required for DNA repair. To explore this possibility further, we characterized nucleotide metabolism, such as the synthesis of nucleotides through the pentose phosphate pathway (PPP) or pyrimidine salvage. To assess if

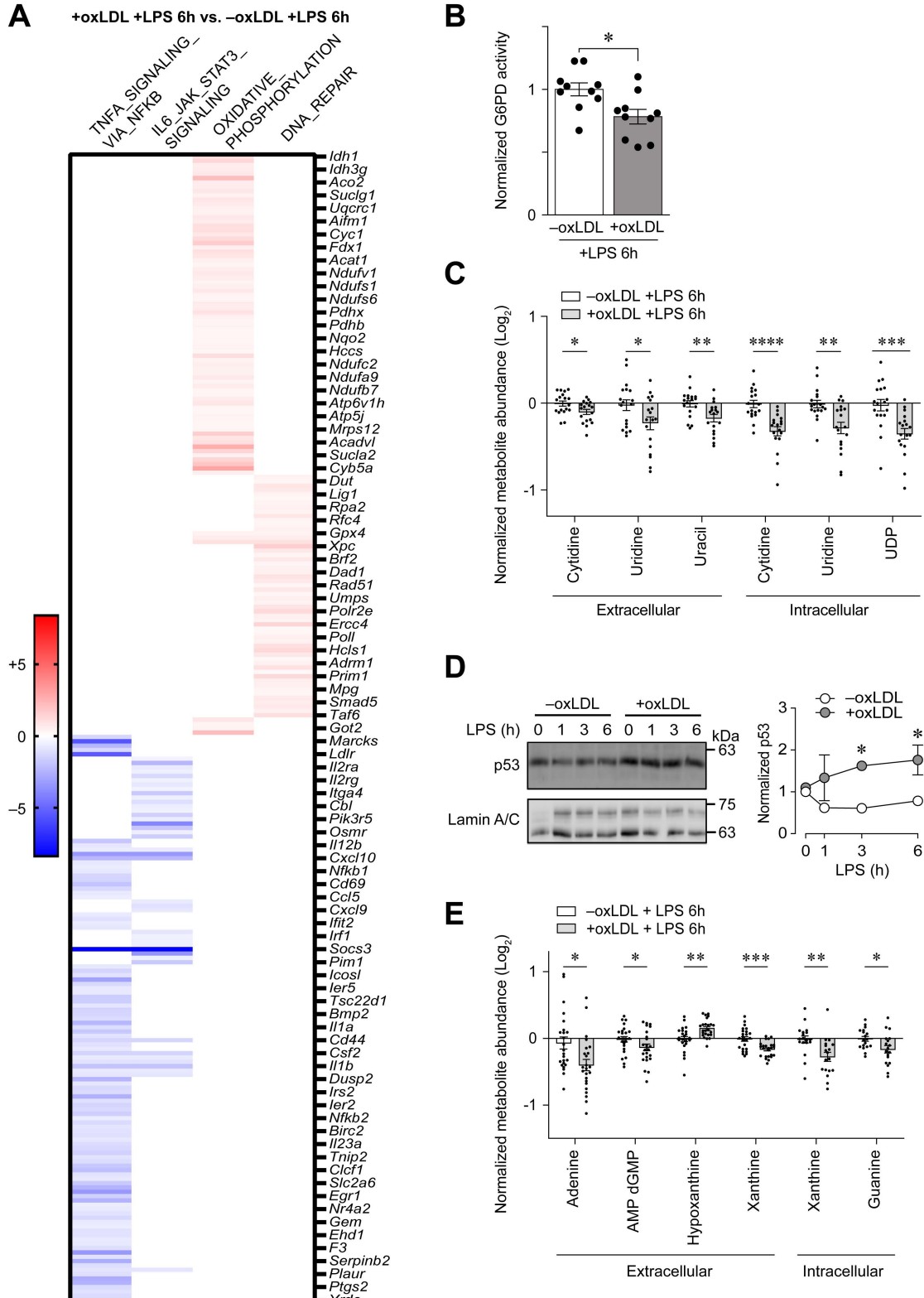

**Fig 1. LPS stimulation of PMφs with accumulated oxLDL in induces DNA damage repair responses and modulates nucleotide metabolism.** (A) Hallmark pathway analysis of bulk RNA-sequencing datasets from cultured PMφs with and without oxLDL accumulation and 6 h of LPS stimulation. The heatmap includes genes that contributed most to the pathway enrichment

and expression values are colored to represent enrichment from low (blue) to high (red). (B) Normalized G6PD activity in PMφs ± oxLDL accumulation and 6 h of LPS stimulation (n = 10). (C) Quantification of extracellular and intracellular metabolites related to pyrimidine salvage synthesis pathways in PMφs ± oxLDL accumulation and 6 h of LPS stimulation. Data were generated by LC-MS metabolomics (n = 18). (D) Representative immunoblots and quantification of nuclear p53 protein expression in PMφs ± oxLDL accumulation. A LPS stimulation time-course (0 – 6 h) was performed. Data are normalized to the corresponding Lamin A/C and the 0 h LPS time point of the –oxLDL group (assigned a value of 1, n = 3). (E) Quantification of extracellular and intracellular metabolites related to purine salvage synthesis pathways in PMφs ± oxLDL accumulation and 6 h of LPS stimulation. Data were generated by LC-MS metabolomics (n = 18). The mean ± SEM is plotted in all graphs. Significant differences were determined by unpaired two-way Student's $t$ tests in (B), (C) and (E), and a two-way ANOVA with Bonferroni post hoc test in (D) ( * P < 0.05, ** P < 0.01, *** P < 0.001, **** P < 0.0001).

the PPP is modulated by oxLDL loading, we first measured the activity of glucose-6-phosphate dehydrogenase (G6PD), the enzyme that catalyzes the rate-limiting step of the PPP. As shown in Fig 1B, oxLDL loading of PMφs suppressed G6PD activity 6 h after LPS stimulation, which implicates that the synthesis of nucleotides through the PPP was reduced. This agrees with our previous findings that oxLDL loading of PMφs significantly inhibited the abundance of metabolites produced by the PPP [10] and suggests that the PPP is unlikely to be involved in supporting the increased demands of nucleotides required for DNA repair.

We next investigated if oxLDL loading of PMφs modulates nucleotide metabolism through salvaging extracellular nucleosides, as well as their purine and pyrimidine derivatives, to support nucleotide synthesis. To assess this, we re-analyzed the metabolomics data that we have previously acquired from oxLDL-loaded PMφs 6 h after LPS stimulation [10]. As shown in Fig 1C, compared to control, oxLDL loading significantly depleted specific extracellular and intracellular nucleosides with pyrimidine derivatives (i.e., uridines and cytidines), pyrimidines (i.e., uracil) and uridine diphosphate (UDP), which can be converted to deoxythymidine triphosphate for DNA synthesis. The effect of oxLDL loading on modulating pyrimidine metabolism is also supported by the increased nuclear levels of p53, a well-known transcription factor that is activated by DNA damage [21] and pyrimidine deficiency [22], as detected by immunoblotting of oxLDL-loaded PMφs post LPS stimulation (Fig 1D). Finally, apart from nucleosides with pyrimidine derivatives, we also found that oxLDL loading significantly modulated specific extracellular and intracellular nucleosides with purine derivatives (i.e., hypoxanthine) and purines (i.e., guanine, xanthine) (Fig 1E). Collectively, these results suggest that LPS stimulation of oxLDL-loaded PMφ significantly modulates nucleotide metabolism and promotes the salvage of extracellular pyrimidines and purines to support DNA repair responses.

We next explored mechanisms initiating the DNA damage repair response. As was shown in Fig 1A, apart from upregulating a DNA repair response, oxLDL loading of PMφs also elevated the expression of genes associated with oxidative metabolism in the mitochondria (OXIDATIVE_PHOSPHORYLATION_ signal) post LPS stimulation. MtROS is known to induce DNA damage and repair [23]; therefore, we speculated that oxLDL loading may enhance mitochondrial oxidative metabolism in PMφs after LPS stimulation and hence contribute to the initiation of DNA repair responses. To evaluate this possibility, we profiled PMφ metabolism, specifically the oxygen consumption rate (OCR), in oxLDL-loaded cells after LPS stimulation using the Seahorse Analyzer. LPS treatment reduced OCR during the first 6 h after LPS stimulation in control PMφs (not loaded with oxLDL) (S1 Fig), a finding that is consistent with previous reports of LPS-induced Warburg effect in Mφs, i.e., a metabolic shift from mitochondrial metabolism to glycolysis [24]. OxLDL loading did not significantly modulate OCR when compared to control PMφs (S1 Fig). This suggests that enhanced mitochondrial metabolism in oxLDL-loaded PMφs is not a major contributor to DNA repair.

## Intracellular accumulation of oxLDL or cholesterol in Mφs enhances PARP1-mediated DNA repair response after LPS stimulation

To acquire further mechanistic insights into the induction of DNA repair responses in oxLDL-loaded PMφs, we measured the activity of ARTD enzymes by immunoblotting with an antibody against poly-ADP-ribosylated proteins. OxLDL accumulation in PMφs significantly enhanced ADP-ribosylation of proteins after LPS stimulation (Fig 2A). A similar effect was observed in cholesterol-loaded PMφs (Fig 2B). To determine if PARP1 was primarily responsible for ADP-ribosylation of proteins, we inhibited PARP1 enzymatic function in cholesterol-loaded and LPS-stimulated (1 h) PMφs with AG-14361, a relatively specific PARP1 inhibitor, and found that AG-14361 pretreatment reduced total levels of ADP-ribosylated proteins in a concentration dependent manner (Fig 2C). This result suggests that PARP1 is the main member of the ARTD family that is responsible for mediating the enhanced levels of ADP-ribosylation in cholesterol-loaded PMφs post LPS stimulation. We also measured the phosphorylation of H2A.X Ser139 (γ-H2A.X) as this is the primary marker associated with double-strand DNA breaks and the recruitment of PARP1 [25]. Immunoblotting showed that cholesterol loading of PMφs significantly enhanced γ-H2A.X abundance after LPS stimulation (Fig 2D). Notably, cholesterol loading alone induced γ-H2A.X even prior to LPS stimulation, which correlates with immunoblotting for ADP-ribosylation of proteins shown in Fig 2B. We also assessed the abundance of γ-H2A.X with immunofluorescent microscopy combined with computational analysis, a more accurate and sensitive way to detect and quantify γ-H2A.X levels. Understanding that the area of γ-H2A.X foci highly correlates with the extent of DNA damage [26], we measured γ-H2A.X foci area in oxLDL- and cholesterol-loaded PMφs prior to and after LPS stimulation. OxLDL loading significantly enhanced the area of γ-H2A.X foci only after LPS stimulation (Fig 2E), whereas cholesterol loading enhanced the area of γ-H2A.X foci before and after LPS stimulation (Fig 2F). This suggests that loading of PMφs with cholesterol, but not oxLDL, causes DNA damage and recruitment of PARP1 prior to LPS stimulation. In contrast, both oxLDL and cholesterol loading induced PARP1-dependent response post LPS stimulation.

## Intracellular accumulation of oxLDL and cholesterol in Mφs enhances PARP1-dependent NAD$^+$ consumption

Apart from pyrimidines and purines, another metabolite that is closely associated with DNA damage is NAD$^+$ due to its consumption by ARTDs [23]. Our previous metabolomics experiments revealed that oxLDL loading of PMφ significantly depleted NAD$^+$ 6 h after LPS stimulation [10]. The mechanism for this was not explored. To test the possibility that an enhanced PARP1-dependent response observed in oxLDL-loaded PMφs accounts for the reduction of NAD$^+$, we first assessed the steady state levels of NAD$^+$ at an earlier time point. Loading of PMφs with oxLDL or cholesterol significantly reduced NAD$^+$ at 3 h after LPS stimulation (Fig 3A). Next, we assessed if loading of PMφs with oxLDL or cholesterol alone could modulate NAD$^+$ metabolism. Interestingly, loading of PMφs with cholesterol, but not oxLDL, significantly reduced NAD$^+$ (Fig 3B), consistent with the data presented in Fig 2. To further explore how cholesterol loading of PMφs modulates NAD$^+$ metabolism, we measured NAD$^+$ during cholesterol loading. Compared to ethanol carrier (-Chol), culture of PMφs with free cholesterol reduced NAD$^+$ in a time-dependent manner, with a significant difference detected as early as 2 h (Fig 3C).

We explored if the consumption of NAD$^+$ is due to PARP1 enzymatic activity, keeping in mind that the steady-state abundance of NAD$^+$ is regulated by both synthesis (i.e., the NAD$^+$ salvage pathway) and consumption (Fig 3D). We devised an assay that measures the rate of

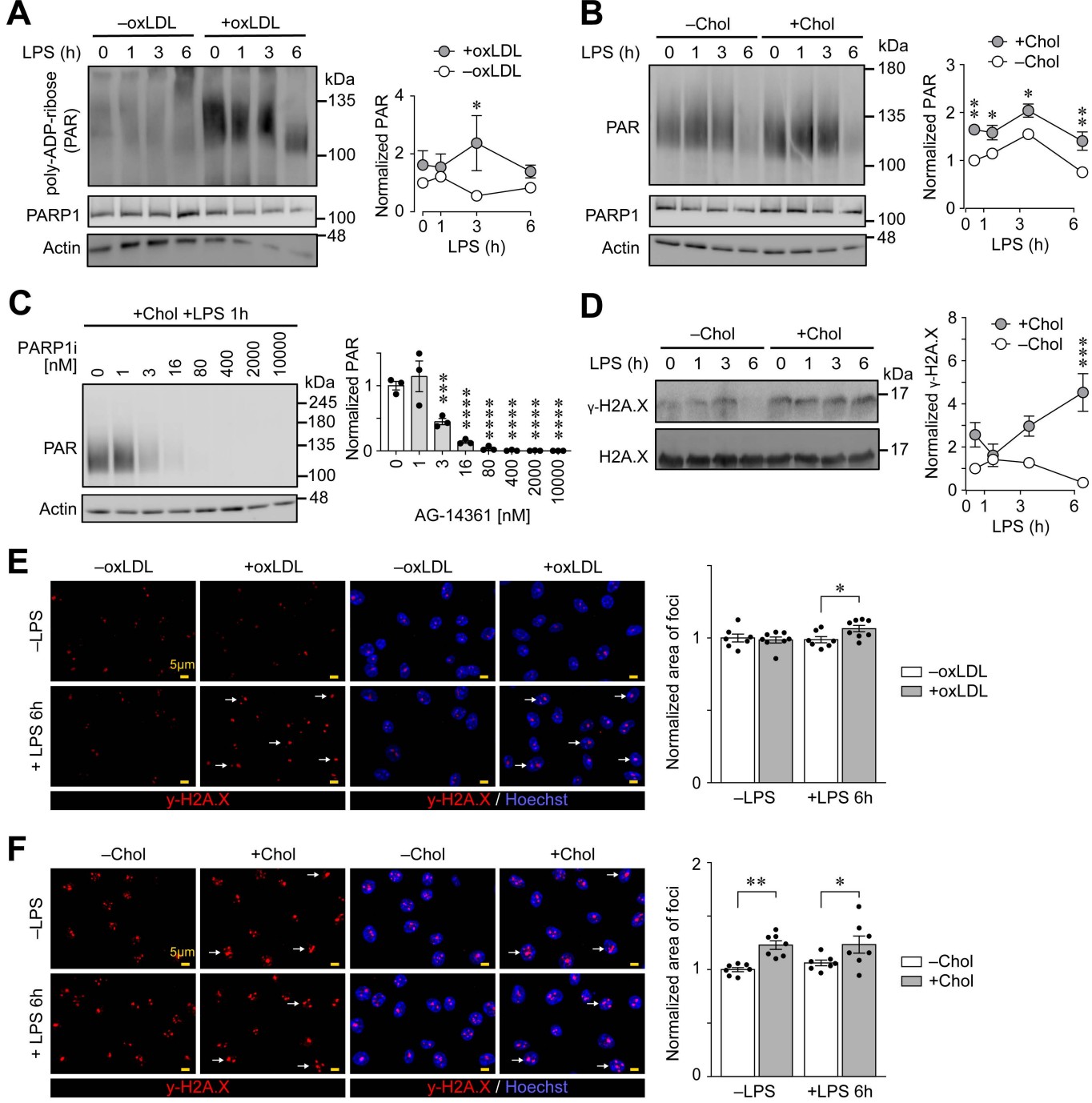

**Fig 2. Loading of PMφs with oxLDL or cholesterol enhances a PARP1 response following stimulation with LPS.** (A and B) Representative immunoblots and quantification of poly-ADP-ribosylated (PAR) proteins in PMφs with or without accumulated oxLDL (A, n = 4) or free cholesterol (B, n = 5). A LPS stimulation time-course (0 – 6h) was performed. Data are normalized to the corresponding actin and the 0h LPS time point of the –oxLDL/Chol groups (assigned a value of 1). (C) Representative immunoblots and quantification of PAR proteins in PMφs loaded with free cholesterol and treated with increasing concentration of PARP1 inhibitor (AG-14361, AG14361). AG14361 was added 1h prior to a 1h LPS stimulation (n = 3). (D) Representative immunoblots and quantification of γ-H2A.X in PMφs with or without accumulated cholesterol. A LPS stimulation time-course (0 – 6h) was performed. Data are normalized to the corresponding H2A.X and the 0h LPS time point of the –Chol group (assigned a value of 1, n = 3). (E and F) Representative fluorescence microscopy images and quantification of PMφs immunostained for γ-H2A.X (red, left panels), and the same images with nuclei counterstained by Hoechst 33342 (blue, right panels). PMφs were cultured for 24h with or without oxLDL (E, n = 7 of 8) or free cholesterol (F n = 7) and indicated cells were stimulated with LPS for 6h. Arrows highlight representative γ-H2A.X foci that are enlarged and are located in nuclei. Scale bars represent 5 μm. The mean ± SEM is plotted in all graphs. Significant differences were determined by a one-way ANOVA and Bonferroni post hoc test (C) or a two-way ANOVA and Bonferroni post hoc test (A, B, D – F) (* P < 0.05, ** P < 0.01, *** P < 0.001, **** P < 0.0001).

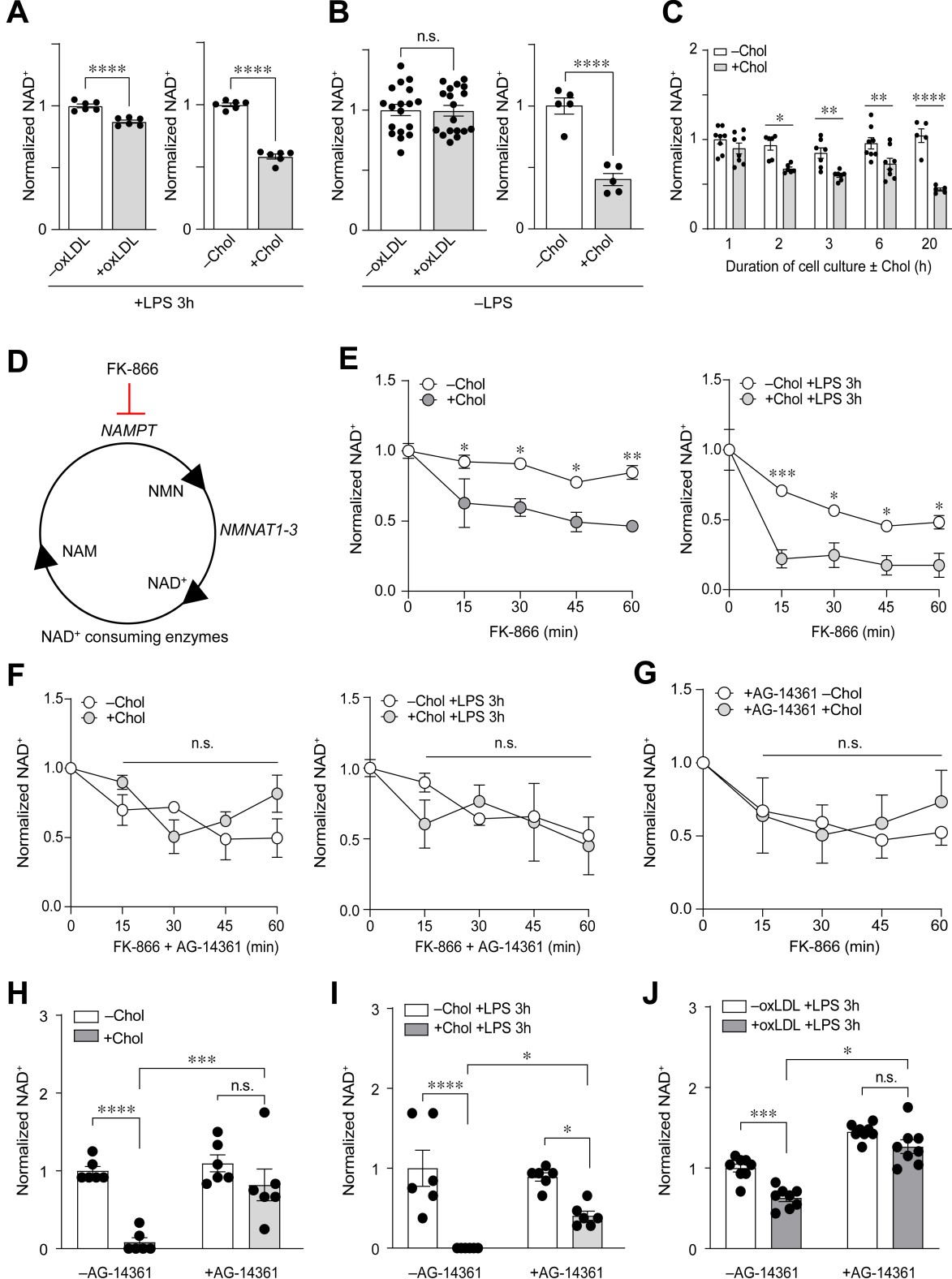

**Fig 3. Loading of PMφs with oxLDL or cholesterol enhances PARP1-dependent NAD⁺ consumption.** (A) Quantification of NAD⁺ abundance in PMφs with or without accumulated oxLDL (n = 6) or free cholesterol (n = 6) and 3 h after LPS stimulation. Data are normalized the −oxLDL/Chol groups (assigned a value of 1). (B) Quantification of NAD⁺ abundance in PMφs with or without accumulated

oxLDL (n = 18) or free cholesterol (n = 5). Cells were not stimulated with LPS. Data are normalized the –oxLDL/Chol groups (assigned a value of 1). (C) Quantification of NAD$^+$ abundance in PMφs at different time points during free cholesterol loading (n = 5-8) Data are normalized to the 0 h LPS time point of the –Chol group (assigned a value of 1). (D) A schematic illustrating the NAD$^+$ salvage pathway. FK866 is a specific inhibitor that blocks the function of NAMPT, a rate-limiting enzyme of the NAD$^+$ salvage pathway. (E - G) Quantification of NAD$^+$ consumption in PMφs by addition of FK-866 every 15 minutes for up to 1 h. For each group, the data are normalized to NAD$^+$ values just prior to the addition of FK-866 (0 min time point, assigned a value of 1). (E) PMφs with or without accumulated free cholesterol and with or without LPS stimulation for 3 h (n = 4). (F) PMφs with or without accumulated cholesterol before and after 3 h of LPS stimulation (n = 3-4). AG-14361 was added with FK-866 every 15 minutes for up to 1 h. (G) PMφs with or without accumulated free cholesterol (n = 3). AG-14361 was added to 1 h prior to culture with free cholesterol or ethanol carrier and after 24 h, FK-866 was added every 15 minutes for up to 1 h. (H - J) Quantification of NAD$^+$ abundance in PMφs with or without accumulated cholesterol (H, I) or oxLDL (J). AG-14361 was added 1h prior to cholesterol or oxLDL loading. LPS stimulation for 3 h was absent in H (n = 6) and present 3 h in I (n = 6) after J (n = 8). Data are normalized the –Chol/oxLDL groups (assigned a value of 1). The mean ± SEM is plotted in all graphs. Significant differences were determined by unpaired two-way Student's *t* test (A – B), or by a two-way ANOVA and Bonferroni post hoc test (E– J) (* P < 0.05, ** P < 0.01, *** P < 0.001, **** P < 0.0001).

NAD$^+$ consumption by acutely inhibiting the function of NAMPT, the rate-limiting enzyme that mediates NAD$^+$ synthesis through the salvage pathway using a relatively specific inhibitor FK-866. Using this assay, we found that cholesterol loading of PMφs significantly enhanced the consumption rate of NAD$^+$ before LPS stimulation and 3 h after (Fig 3E). To determine if the enhanced NAD$^+$ consumption rate is due to PARP1, we repeated the FK-866 assay with two different PARP1 inhibitors: AG-14361 and rucaparib. Upon inhibition of PARP1, cholesterol loading of PMφs no longer accelerated the consumption rate of NAD$^+$ either before or after LPS stimulation (Fig 3F, S2A and S2B Fig). We also observed a reduction of NAD$^+$ consumption when PARP1 was inhibited prior to cholesterol loading (Fig 3G). PARP1 inhibition did not modulate the induction of ABCA1 protein by cholesterol loading (S2C Fig). Taken together, these findings demonstrate that cholesterol loading of PMφs enhanced PARP1-dependent NAD$^+$ consumption both before and after LPS stimulation.

Finally, we investigated if blocking PARP1 enzymatic function could restore the steady-state NAD$^+$ levels in PMφs loaded with oxLDL or cholesterol. Upon PARP1 inhibition, NAD$^+$ abundance in cholesterol-loaded PMφs was comparable to control cells prior to LPS stimulation (Fig 3H); however, in LPS-stimulated PMφs cholesterol loading still suppressed NAD$^+$ levels, although to a lesser extent when compared to cells without PARP1 inhibition (Fig 3I). Inhibition of PARP1 enzymatic function restored NAD$^+$ steady state abundance in oxLDL-loaded PMφs 3 h after LPS stimulation (Fig 3J). Overall, these findings suggest that both cholesterol and oxLDL loading-induce depletion of NAD$^+$ upon LPS stimulation, which is primarily due to activation of PARP1, although additional mechanisms are likely to be involved in cholesterol-loaded PMφs.

## Mitochondrial ROS induced by cholesterol loading mediates the PARP1-dependent response

Our findings that cholesterol accumulation in PMφs prior to LPS stimulation increases ADP-ribosylation of proteins (Fig 2B and 2C), γ-H2A.X (Fig 2F) and consumption of NAD$^+$ (Fig 3F–H) in a PARP1-dependent manner, suggest that cholesterol accumulation in PMφs alone can induce DNA damage and thus activate a PARP1-dependent DNA repair response. Previously we showed that cholesterol loading of PMφs could induce mtROS in a time-dependent manner [12]. Given that mtROS is linked to PARP1-dependent DNA damage response [23], we explored the role of cholesterol loading-induced mtROS in activating a PARP1 response. We performed mitoSOX staining of cholesterol-loaded PMφs and confirmed that cholesterol loading induced mtROS and that treatment during cholesterol

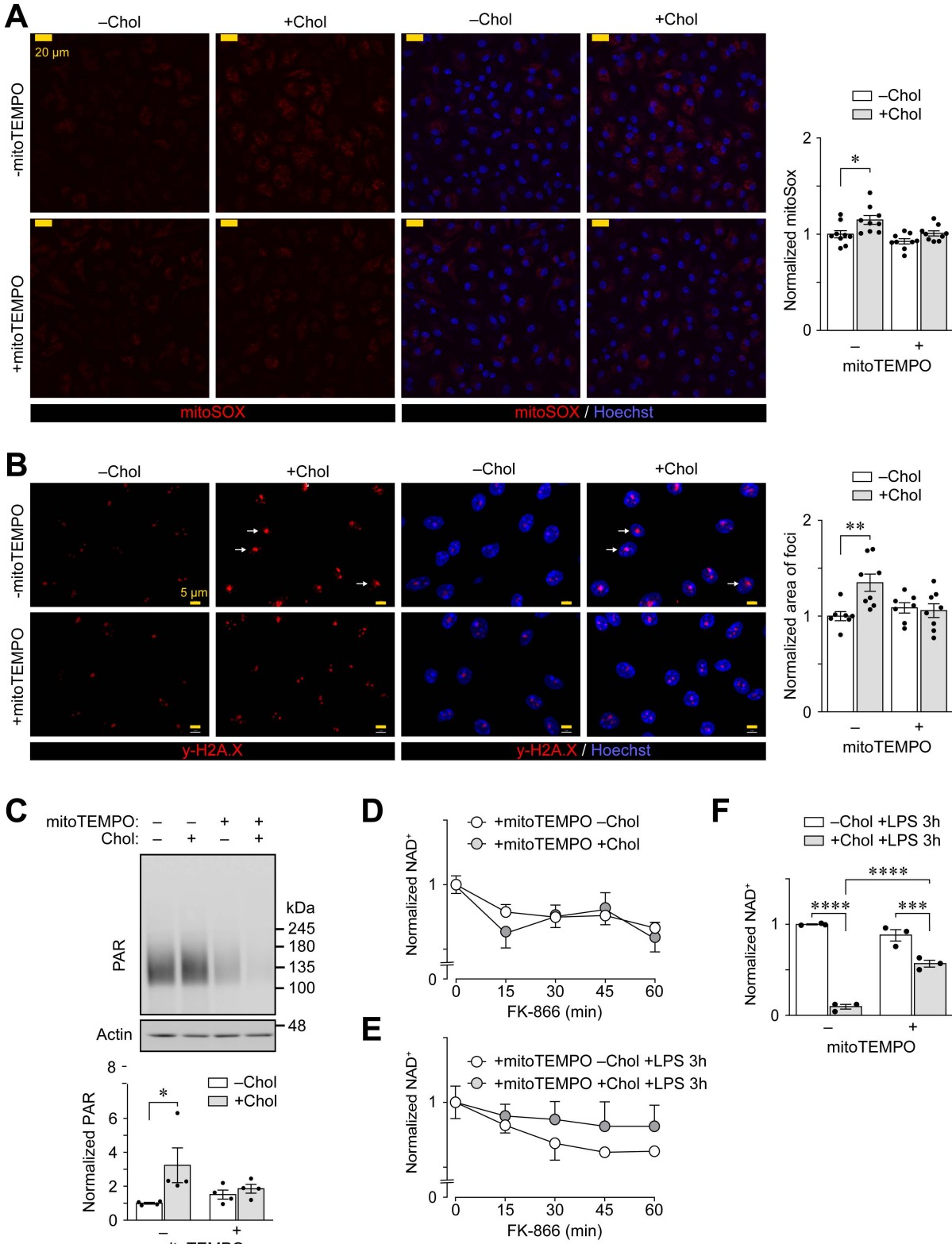

**Fig 4. Mitochondrial ROS induced by cholesterol loading mediates the PARP1-dependent response.** (A) Representative fluorescence microscopy images and quantification of mtROS. PMφs were cultured with or without mitoTEMPO and cholesterol (Chol), as indicated. In this and subsequent experiments, mitoTEMPO was added 1 h prior to culture with free cholesterol for 24 h. Staining with mitoSOX (red,

left panels) and the same images with nuclei counterstained by Hoechst 33342 (blue, right panels). Scale bars represent 20 μm (n = 9). (B) Representative fluorescence microscopy images and quantification of PMφs immunostained for γ-H2A.X (red, left panels) and the same images with nuclei counterstained by Hoechst 33342 (blue, right panels). PMφs were cultured with or without mitoTEMPO and cholesterol, as indicated. Arrows highlight representative enlarged γ-H2A.X foci in nuclei. Scale bars represent 5 μm (n = 7-8). (C) Representative immunoblots and quantification of poly-ADP-ribosylated (PAR) proteins in PMφs cultured with or without mitoTEMPO and cholesterol, as indicated. Data are normalized to the corresponding actin and the –Chol group (assigned a value of 1, n = 4). (D and E) Quantification of NAD$^+$ consumption. PMφs were cultured with or without mitoTEMPO and cholesterol, as indicated. Experiments were performed before (D, n = 6) and 3 h after LPS stimulation (E, n = 4). FK-866 was added every 15 minutes for up to 1 h. (F) Quantification of NAD$^+$ abundance in PMφs cultured with or without mitoTEMPO and cholesterol, as indicated, then stimulated with LPS for 3 h (n = 6). The mean ± SEM is plotted in all graphs. Significant differences in were determined using a two-way ANOVA and Bonferroni post hoc test ( * P < 0.05, ** P < 0.01, *** P < 0.001, **** P < 0.0001).

loading with mitoTEMPO, a specific scavenger of mtROS, blunted this response (Fig 4A). Treatment of PMφs with mitoTEMPO also inhibited the induction of γ-H2A.X (Fig 4B) and ADP-ribosylation of endogenous proteins (Fig 4C). The enhanced rate of NAD$^+$ consumption in cholesterol-loaded PMφs (Fig 3E) was abrogated by mitoTEMPO in cells without (Fig 4D) or with LPS stimulation for 3 h (Fig 4E). MitoTEMPO partially reversed the dramatic lowering of steady state NAD$^+$ in cholesterol-loaded PMφs 3 h after LPS stimulation (Fig 4F) and the effect was similar to inhibition of PARP1 (Fig 3I), suggesting that other mechanisms, in addition to PARP1-dependent consumption of NAD$^+$ levels, are likely to be involved in regulating NAD$^+$ metabolism. Overall, our findings demonstrate that cholesterol accumulation in PMφs induces mtROS, which mediates a PARP1-dependent response.

## Mitochondrial ROS partially contributes to the suppression of LPS-induced inflammation in PMφs with accumulated cholesterol

ROS induced by lipid loading has been shown to reprogram trained immunity of oxLDL-treated monocytes [27]. Whether ROS can modulate Mφ inflammatory responses has not been explored. We therefore cultured PMφs with mitoTEMPO 1 h prior to loading with oxLDL or cholesterol. Treatment with mitoTEMPO during loading with oxLDL did not alter the suppression of a LPS-induced inflammatory response (Fig 5A). In contrast, treatment of PMφs with mitoTEMPO during cholesterol loading partially restored LPS-induced inflammatory responses (Fig 5B). We next investigated how the induction of mtROS reprograms the inflammatory response of cholesterol-loaded PMφs. Given that mtROS can stabilize NRF2 [28] and the upregulation of NRF2 partially mediates the suppression of inflammatory responses in cholesterol-loaded PMφs [12], we investigated if the rescue effect of mtROS is NRF2-dependent. We performed qPCR to assess the expression of NRF2-dependent genes during cholesterol loading. As expected, based on our previous findings that NRF2 is stabilized during cholesterol loading [12], loading of PMφs with cholesterol significantly up-regulated the expression of NRF2-dependent genes (Fig 5C). However, inhibition of mtROS production with either diphenyleneiodonium, which blocks ROS production by NADPH oxidases and mitochondria, or mitoTEMPO did not reduce the expression of NRF2-dependent genes induced by cholesterol loading (Fig 5C). These results show that activation of NRF2 remains intact even when the production mtROS is inhibited and suggest that the partial inhibiion of inflammatory gene expression (Fig 5B) is mediated by different pathways.

We next investigated the possibility that mitoTEMPO functions by inhibiting the PARP1 response in cholesterol-loaded PMφs. We inhibited PARP1 enzymatic activity in PMφs during cholesterol loading and found that the expression of LPS-induced inflammatory genes remained suppressed (S3A Fig). We obtained similar data when studying PMφs loaded with oxLDL (S3B Fig).

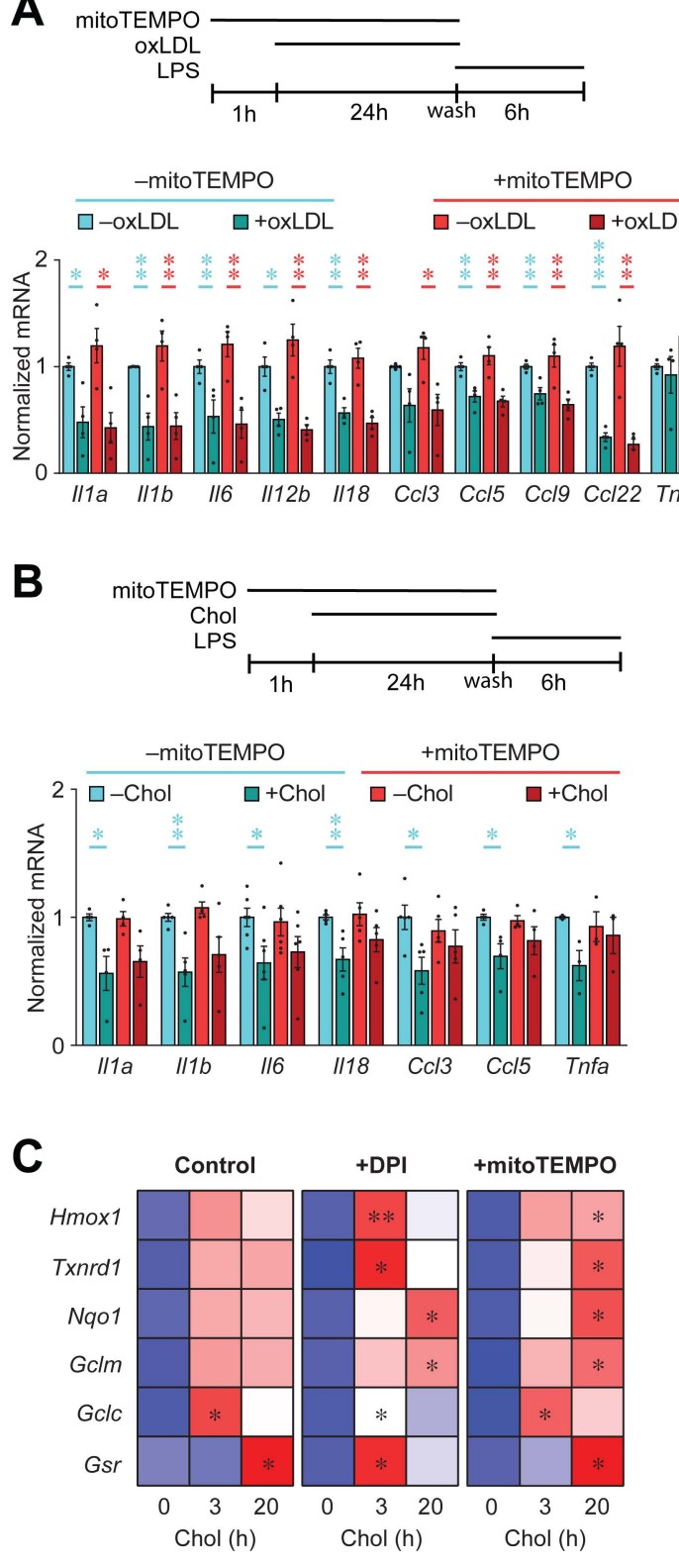

**Fig 5. Mitochondrial ROS partially contributes to the suppression of LPS-induced inflammation in PM φs with accumulated cholesterol.** (A and B) Analysis of inflammatory gene expression by qPCR. MitoTEMPO was added 1 h prior to culturing PMφs with or without oxLDL (A) or free cholesterol (Chol, B). Subsequently PMφs were stimulated

with LPS for 6 h, as is shown in a schematic above each graph. The mean ± SEM is plotted (n = 4). Significant differences in were determined by an unpaired Student's $t$ test (* P < 0.05, ** P < 0.01, *** P < 0.001). (C) Heatmap showing that the induction of NRF2-regulated gene expression by cholesterol loading was not altered by the inhibition ROS. Diphenyleneiodonium (DPI) and mitoTEMPO were added 1 h prior to culturing PMφs with free cholesterol for 0, 3 or 20 h. Gene expression was determined by qPCR (n = 4). In each group, significant differences from the 0 h time point were determined by a one-way ANOVA and Bonferroni post hoc test (* P < 0.05, ** P < 0.01).

## PARP1 suppresses LPS-induced inflammation

We performed siRNA knockdown experiments in RAW264.7 Mφs to suppress the expression of PARP1. RAW cells treated with pooled siRNA sequences targeting PARP1 (siPARP1) showed significantly reduced PARP1 steady state mRNA and protein expression relative to RAW cells treated with scrambled siRNA (siControl) (Fig 6A and 6B). Knockdown of PARP1 with siRNA restored the steady state abundance of NAD$^+$ in cholesterol-loaded RAW264.7 Mφs to levels found in control cells (Fig 6C), similar to the inhibition of PARP1 in PMφs prior to LPS stimulation (Fig 3H), This was also the case, in LPS-stimulated RAW264.7 Mφs with PARP1 knockdown (Fig 6D); however, because the abundance of NAD$^+$ was elevated in LPS-stimulated RAW264.7 Mφs without cholesterol loading, a statistically significant difference remained between LPS-stimulated cells without versus with cholesterol loading (Fig 6D).

We next investigated the effect of PARP1 siRNA knockdown on the induction of inflammatory genes by LPS. SiRNA-treated RAW264.7 Mφs with and without cholesterol loading were stimulated with LPS for 6 h and the expression of inflammatory genes was determined by qPCR. Cholesterol loading of RAW264.7 Mφs impaired the induction of inflammatory genes in cells treated with either scrambled or PARP1 siRNA (Fig 6E). However, in RAW cells without cholesterol loading, PARP1 siRNA knockdown increased *Il1a* and *Il1b* expression significantly, and a similar trend was found for other inflammatory genes (Fig 6E). Increased *Il1a* and *Il1b* expression in RAW cells treated with PARP1 siRNA was not observed in PMφs treated with mitoTEMPO (Fig 5B) or PARP1 inhibitor (S3A Fig) and likely accounted for significant differences between values in the –Chol and +Chol groups. Note that *Il1a* and *Il1b* expression was comparable between the siControl -Chol and siPARP1 +Chol groups (Fig 6E). Collectively data in S3A Fig and Fig 6E suggest that PARP1 dampens LPS-induced inflammatory gene expression through a non-enzymatic function.

## Discussion

The formation of foam cells and their inflammatory responses has always been a topic of controversy. To explore the relationship between lipid loading of Mφs and their inflammatory responses, we and others have cultured primary murine Mφs with oxLDL or cholesterol to model foam cell formation *in vivo* and have consistently reported that lipid loading of Mφs alone is insufficient to drive inflammation [9–12]. Recently, our published studies have demonstrated that oxLDL and cholesterol loading of Mφs impaired their inflammatory response to LPS stimulation, in part by suppressing the induction of glycolysis [10]. Yet, how other metabolic processes are affected and if they contribute to the impairment of inflammatory gene expression remains to be determined. In this study, we found that loading of PMφs with oxLDL or cholesterol enhanced a LPS-induced PARP1-dependent response. This was linked to changes in intracellular metabolic pathways, including an increase of nucleotide synthesis through salvaging pathways, NAD$^+$ consumption, as well as increased levels of γ-H2A.X. Furthermore, we have also discovered that cholesterol loading of PMφs alone was sufficient to induce a PARP1 response prior to LPS stimulation. This was attributed to the production of mtROS during cholesterol loading.

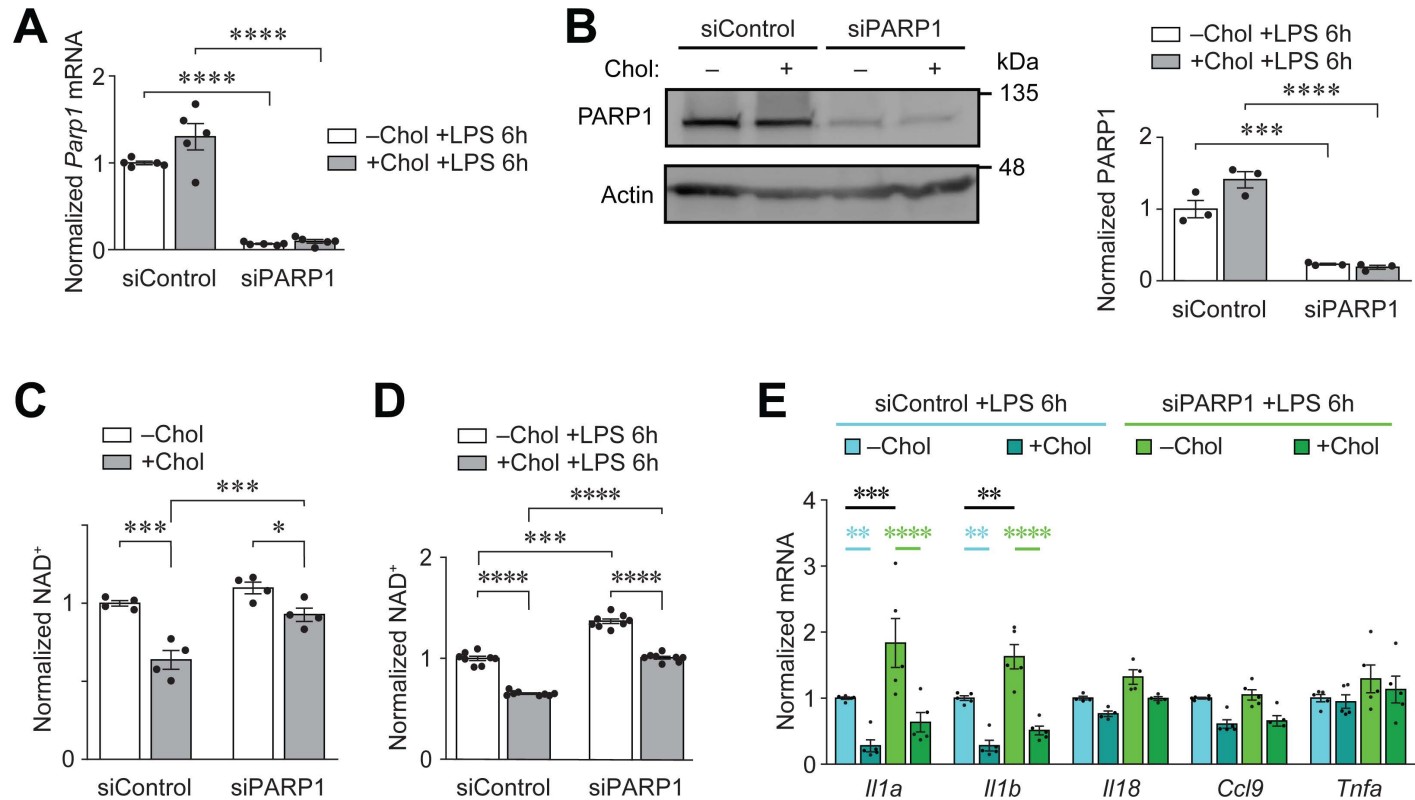

**Fig 6. PARP1 suppresses LPS-induced inflammation.** (A, B) Characterization of PARP1 siRNA knockdown in RAW 264.7 Mφs. *Parp1* mRNA expression detected by qPCR (A, n = 5) and protein expression detected by immunoblotting (B, n = 3). RAW 264.7 Mφs were transfected with scrambled siRNA (siControl) or pooled siRNAs against PARP1 (siPARP1) then were cultured with or without free cholesterol for 24 h and stimulated with LPS for 6 h. (C, D) Quantification of NAD$^+$ abundance in RAW 264.7 Mφs transfected with siControl or siPARP1. Cholesterol loading and LPS stimulation are indicated (no LPS in C, n = 4; and 6 h of LPS stimulation in D, n = 8). (E) qPCR analysis of inflammatory gene expression in RAW 264.7 Mφs transfected with siControl or siPARP1. Cholesterol loading and LPS stimulation are indicated (n = 5). The mean ± SEM is plotted in all graphs. Significant differences were determined by a two-way ANOVA and a Bonferroni post hoc test (** P < 0.01, *** P < 0.001, **** P < 0.0001).

An important theme that emerges from our current study, along with others [11,27], is the notion that lipid loading-induced mtROS can modulate the inflammatory responses of myeloid cells upon their activation. Indeed, we have previously measured mtROS levels and found that lipid loading of PMφs with either oxLDL or cholesterol can induce mtROS during lipid loading [10,11]. Notably, alleviating oxLDL loading-induced mtROS reinstated the acetylation of H3K27 levels on a subset of LPS-induced pro-inflammatory genes [11]. Similarly, in this study, we found that alleviating mtROS during cholesterol loading blocks the PARP1 response (Fig 4B) and partially restores LPS-induced inflammatory gene expression (Fig 5B). A key question that remains to be answered is the mechanism behind the induction of mtROS during cholesterol loading of PMφs. Our past study has shown that cholesterol loading of PMφs activated the production of mtROS in a time-dependent manner (with a peak at 12 h), suggesting that this induction of mtROS is likely a metabolic consequence for adapting to the accumulation of excess intracellular cholesterol, rather than an inflammatory response downstream of receptor-mediated signaling cascades [12]. Indeed, oxLDL loading-induced production of mtROS is also time-dependent and is associated with the upregulation of fatty acid oxidation and mitochondrial metabolism [11] but is not associated with a PARP1 response (Fig 2E) and mitoTEMPO inhibition of mtROS during oxLDL loading did not restore LPS induced inflammatory gene expression (Fig 5A).

Incubation with oxLDL or free cholesterol is a reproducible method to induce the formation of lipid droplets in Mφs [8,9], enabling us to model certain aspects of atherosclerotic-derived Mφ foam cells in an *in vitro* setting [10]. In this study, we have characterized the inflammatory and metabolic profiles of PMφs loaded with oxLDL or cholesterol after LPS stimulation. While we found that they both generally displayed similar functional responses, several key biological differences have also been observed, one of which is their intrinsic ability to activate PARP1. Throughout the study, we have consistently observed that cholesterol loading of PMφs alone (without LPS stimulation) is sufficient to induce PARP1 activation, as measured by the induction of ADP-ribosylated proteins, $NAD^+$ consumption and γ-H2A.X increase. On the other hand, the induction of a PARP1 response was not reproduced by oxLDL loading of PMφs, which is surprising as oxLDL has been shown to induce DNA damage in endothelial cells [29,30] and we have previously shown that oxLDL loading of PMφs could induce mtROS production [10]. Therefore, it remains unclear how oxLDL loading of PMφs could induce an enhanced PARP1-dependent response after LPS stimulation (Fig 2A). On the other hand, the ability of cholesterol loading to intrinsically induce γ-H2A.X may underlie enhancement of the PARP1 response in PMφs after LPS stimulation. For instance, recent research has shown that γ-H2A.X serves more than just a site that marks the enrichment of PARP1 [25]. Specifically, the authors have characterized H2A and γ-H2A.X nucleosomes and found that γ-H2A.X directly promotes PARP1 stabilization and catalysis [25]. Therefore, in the context of our study, it is likely that cholesterol loading-induced γ-H2A.X primed the stabilization of PARP1 and catalysis prior to LPS stimulation, enabling them to be more enzymatically efficient upon LPS stimulation and thus leading to the observed enhanced PARP1-dependent response (Fig 2B). Interestingly, our previous study showed that cholesterol loading of Mφs primed NRF2 stabilization by disrupting the interaction of NRF2 with KEAP1 cysteine residues [12]. In cholesterol-loaded Mφs, stimulation with LPS further enhanced NRF2 stabilization and impaired HIF-1α-dependent inflammation and glycolysis [12]. Collectively, our current and previous findings suggest that cholesterol loading of Mφ can suppress LPS-induced inflammation by pre-activating multiple cytoprotective pathways (i.e., NRF2, PARP1) even prior to LPS stimulation.

Apart from this, another key difference that was observed between oxLDL and cholesterol loading of PMφs is the role of PARP1-dependent consumption of $NAD^+$. For instance, as is shown in Fig 3J, inhibiting the function of PARP1 fully restored $NAD^+$ steady-state abundance in oxLDL-loaded PMφs, yet this only partially restored $NAD^+$ abundance in cholesterol-loaded PMφs (Fig 3I). While this suggests that both oxLDL and cholesterol loading-induced depletion of $NAD^+$ levels is due to the enhanced consumption of $NAD^+$ by PARP1, additional mechanisms are likely to be involved in cholesterol-loaded PMφs, such as an impairment of $NAD^+$ synthesis. In fact, this may also explain why the reduction of $NAD^+$ levels mediated by cholesterol loading was far greater in magnitude relative to oxLDL loading (Fig 3A).

PARP1 is best known for its role in mediating repair of DNA damage, yet its role in orchestrating other processes, such as lipid metabolism, has only been recently revealed [31]. For instance, it was reported that while lipid-associated molecules can activate PARP1 responses, its inhibition could enhance cholesterol efflux in macrophages through ABCA1 in a LXR-dependent manner [32]. Thus, this implicates that PARP1 may play a pivotal role in regulating cholesterol levels in foam cells. However, in this study, we found that inhibition of PARP1 did not modulate the increased expression of ABCA1 protein induced by cholesterol loading (S2C Fig). Similarly, another study has also found that pharmacological inhibition of PARP1 with thieno[2,3-c]isoquinolin-5-one (TIQ-A) did not affect ABCA1 protein expression [33].

While the role of PARP1 in mediating lipid metabolism through ABCA1 remains controversial, the role of PARP1 in mediating inflammation is relatively well-established [18]. This

is significant as it raises the possibility that PARP1 may repair DNA damage at the expense of its role in mediating inflammatory processes, and thus this may be a novel mechanism to fine tune the extent of inflammation. To evaluate this possibility, we characterized the optimal concentration of a specific inhibitor of PARP1 enzymatic function (Fig 2C) and assessed if it can modulate Mφ inflammatory gene expression in response to LPS stimulation. However, pharmacological blockade PARP1 enzymatic activity failed to modulate LPS-induced inflammatory gene expression in both control and lipid-loaded (oxLDL or cholesterol) PMφs (S3A and S3B Fig), demonstrating that the DNA repair activity of PARP1 does not contribute significantly to the regulation LPS-induced inflammation. This result is surprising based on reports of PARP1's role in regulating the transcription of inflammatory genes. Interestingly, it has been reported that the impairment of NF-κB-dependent transcriptional activation in fibroblasts with genetic deficiency of *Parp1* was not reproducible with specific PARP1 inhibitors [16]. The discrepancy of inflammatory responses between the genetic deletion of *Parp1* and pharmacological inhibition motivated us to compare our PARP1 inhibitor data with PARP1 siRNA knockdown. Indeed, inhibiting the expression of PARP1 with siRNA in RAW264.7 Mφs enhanced LPS-induced expression of inflammatory genes irrespective of cholesterol loading (Fig 6E). This effect that was not observed with the use of the PARP1 inhibitor or inhibition of mtROS. Overall, these results suggest that independent of its catalytic activity, PARP1 may perform "moonlighting" functions to inhibit inflammatory gene expression in response to stimulation of Toll-like receptor 4 by LPS. For example, PARP1 can negatively regulate glycolysis, e.g., $NAD^+$ depletion [34] and inhibition of hexokinase 1 [35]. The induction of glycolysis is critical for optimal Mφ inflammatory responses, which raises the possibility that the overactivation of PARP1 activity in cholesterol-loaded Mφs may contribute to suppression of glycolysis observed in our previous study [12]. However, how PARP1 mechanistically functions as a negative regulator of Mφ inflammatory responses independent of its DNA repair activity and genome stability maintenance remains a topic for future studies.

## Supporting information

**S1 Fig. Characterizing the effect of oxLDL loading on the metabolic profile of PMφs after LPS stimulation.** Real-time OCR measurements were performed using a Seahorse analyzer and PMφs with or without oxLDL accumulation (±oxLDL) and LPS stimulation. OCR measurements were normalized to one reading cycle prior to LPS injection (indicated by an arrow). The mean ± SEM is plotted (n = 4). Significant differences between corresponding time points in −oxLDL versus +oxLDL groups were not detected using a one-way ANOVA and a Bonferroni post hoc test.
(PDF)

**S2 Fig. Characterizing the effect of rucaparib on cholesterol-loaded PMφs.** (A, B) Quantification of NAD+ consumption in PMφs. FK-866, a NAMPT inhibitor, and rucaparib, a PARP1 inhibitor, were added every 15 minutes for up to 1 h. For each group, the data are normalized to $NAD^+$ values just prior to the addition of FK-866 and rucaparib (0 min time point, assigned a value of 1). (A) PMφs with or without accumulated cholesterol without LPS stimulation (n = 3). (B) PMφs with or without accumulated cholesterol and 3 h after LPS stimulation (n = 3). (C) ABCA1 protein expression detected by immunoblotting PMφs with or without free cholesterol loading (n = 3). Rucaparib or DMSO control was added 1 h prior to culturing cells with or without cholesterol for 24 h. The ABCA1 values are normalized to corresponding actin and the −Rupacarib and −Chol group (assigned a value of 1). The mean ± SEM is plotted in all graphs. Significant differences were determined by a two-way ANOVA and a Bonferroni post hoc test (* P < 0.05).
(PDF)

**S3 Fig. Inhibition of PARP1 enzymatic activity does not rescue the suppression of LPS-induced inflammatory gene expression in lipid-loaded PMφs.** (A, B) AG14361, a PARP1 inhibitor, was added 1 h prior to culturing PMφs with or without free cholesterol (Chol, A, n = 3-5) or oxLDL (B, n = 3–5). Subsequently PMφs were stimulated with LPS for 6 h, as is shown in a schematic above each graph. Inflammatory gene expression was measured by qPCR. The mean ± SEM is plotted (n = 3). Significant differences were determined using an unpaired Student's *t* test (* P < 0.05, ** P < 0.01, *** P < 0.001, **** P < 0.0001).
(PDF)

**S4 Fig. Original uncropped immunoblots.**
(PDF)

## Author contributions

**Conceptualization:** Kenneth K.Y. Ting, Myron I. Cybulsky.

**Data curation:** Kenneth K.Y. Ting.

**Formal analysis:** Kenneth K.Y. Ting, Hisham M. Ibrahim.

**Funding acquisition:** Kenneth K.Y. Ting, Myron I. Cybulsky.

**Investigation:** Kenneth K.Y. Ting, Hisham M. Ibrahim, Nitya Gulati, Yufeng Wang.

**Methodology:** Kenneth K.Y. Ting.

**Project administration:** Kenneth K.Y. Ting.

**Software:** Kenneth K.Y. Ting.

**Supervision:** Jonathan V. Rocheleau, Myron I. Cybulsky.

**Writing – original draft:** Kenneth K.Y. Ting.

**Writing – review & editing:** Kenneth K.Y. Ting, Myron I. Cybulsky.

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
