## [Decision Letter · Decision Letter 0]

7 Aug 2024

PONE-D-24-28324Intracellular accumulation of free cholesterol in macrophages triggers a PARP1 response to DNA damage and PARP1 impairs lipopolysaccharide-induced inflammatory responsePLOS ONE

Dear Dr. Cybulsky,

Thank you for submitting your manuscript to PLOS ONE. After careful consideration, we feel that it has merit but does not fully meet PLOS ONE’s publication criteria as it currently stands. Therefore, we invite you to submit a revised version of the manuscript that addresses the points raised during the review process.

We look forward to receiving your revised manuscript.

Kind regards,

Zhiming Li, Ph.D.

Academic Editor

PLOS ONE

Journal Requirements:

   "This study was supported primarily by the Canadian Institutes of Health Research (CIHR) grant FDN-154299 (M.I.C.). M.I.C. holds a Tier 1 Canada Research Chair. K.K.Y.T was supported by fellowships from Canadian Institutes of Health Research, Ontario Graduate Scholarship, fellowships from the University of Toronto and The Peterborough K. M. Hunter Charitable Foundation."

5. We note that your Data Availability Statement is currently as follows: All relevant data are within the manuscript and its Supporting Information files.

Reviewers' comments:

Reviewer's Responses to Questions

**Comments to the Author**

1. Is the manuscript technically sound, and do the data support the conclusions?

Reviewer #1: Partly

2. Has the statistical analysis been performed appropriately and rigorously? 

Reviewer #1: I Don't Know

3. Have the authors made all data underlying the findings in their manuscript fully available?

Reviewer #1: Yes

4. Is the manuscript presented in an intelligible fashion and written in standard English?

Reviewer #1: Yes

5. Review Comments to the Author

Reviewer #1: Ting et al. evidence a pathway stemming from oxLDL/cholesterol-induced metabolic rearrangements yielding reactive species from the mitochondria leading to PARP activation and proinflammatory cytokine production. The study is very relevant from the perspective of foam cell formation and atherosclerosis. The paper needs to be amended as follows:

Please do not resolve “PARP”. See PMID: 34323016.

The interactions between cholesterol and PARP enzmyes is more complex, please see PMID: 34450194. Please amend the discussion accordingly.

Please report (RT)-qPCR reactions according to PMID: 19246619, PMID: 32746458.

AG-14361 is not the available best inhibitor. Please repeat key experiments with rucaparib or talazoparib.

I am not sure that the inhibition of NAMPT alone is sufficient to block NAD+ salvage, NMNAT blockade would be also necessary.

Does PARP inhibition modulate the expression of ABCA1, in other words, is there a difference in cholesterol efflux from cells?

The authors introduce HIF1 as a key pathway for the metabolic switch in macrophages, later shown NRF2 activation. HIF1 and NRF2 systems are interconnected. that should be discussed.

The PAR blots should be shown as a whole with MW standards. These blots are either cut/cropped or are not showing the usual PARylation pattern. This a key point that needs to be cleared.

Please note that pharmacological PARP inhibitors block both PARP1 and PARP2, hence only by pharmacology it is not possible to distinguish between them. PARP2 is also associated with inflammation and cholesterol metabolism (PMID: 38698556). Please either introduce genetic studies or reshape the discussion. Furthermore, PARPs do modulate metabolism, hence the glycolytic switch may be related to PARP activation. This needs to be at least discussed.

Typo:

line 215 “purines and pyrimidines derivatives” correctly: purine and pyrimidine derivatives.

line 335 Please add the missing reference.

6. PLOS authors have the option to publish the peer review history of their article (what does this mean? ). If published, this will include your full peer review and any attached files.

**Do you want your identity to be public for this peer review?** For information about this choice, including consent withdrawal, please see our Privacy Policy .

Reviewer #1: No

---

## [Author Response · Author response to Decision Letter 1]

7 Jan 2025

RESPONSE TO THE REVIEW

We would like to thank the reviewer for the insightful and helpful comments. Each point raised by the reviewer is in bold font and our response is below. Changes to the text in the revised manuscript are highlighted in yellow.

Please do not resolve „PARP”. See PMID: 34323016.

We would like to thank the reviewer for pointing out the correct nomenclature for mammalian ADP-ribosyltransferases. We now use the term diphtheria toxin-like ADP-ribosyltransferase (ARTD) family.

The interactions between cholesterol and PARP enzymes is more complex, please see PMID: 34450194. Please amend the discussion accordingly.

Thank you for highlighting the complex interaction between cholesterol and PARP enzymes. Indeed, as discussed in the review recommended by the reviewer (PMID: 34450194), various lipid-associated molecules are linked to the activation of PARP1. PARP1 inhibition has also been reported to enhance cholesterol efflux in macrophages through ABCA1 in a LXR-dependent manner, thus implicating that PARP1 may play a pivotal role in regulating cholesterol levels and inflammation in foam cells. As part of our revision, we addressed this experimentally (new Supplementary Figure 2A and 2B) and included a discussion of these results in the revised manuscript.

Please report (RT)-qPCR reactions according to PMID: 19246619, PMID: 32746458.

Thank you for pointing out these publications. Based on section 7 of PMID 19246619, we have provided additional information in the methods section of the revised manuscript, including databased accession numbers of each target and reference gene, the exon locations of each primer, and details about qPCR reactions, such as volumes and primer concentration.

AG-14361 is not the available best inhibitor. Please repeat key experiments with rucaparib or talazoparib.

We have repeated the experiment shown in Figure 3F with rucaparib. These data are presented in Supplementary Figure 2A and 2B. The data suggest that the enhanced consumption of NAD+ in cholesterol loaded macrophages is PARP1-dependent. This is reproducible with or without LPS stimulation. Other experiments in which AG-14361 was used, such as Figure 3H and 3I, were repeated with siRNA against PARP1, as shown in Figure 6. We consider the siRNA approach more specific than pharmacological inhibition.

I am not sure that the inhibition of NAMPT alone is sufficient to block NAD+ salvage, NMNAT blockade would be also necessary.

The reviewer raises an interesting point. Our data show that NAMPT blockade is sufficient to block NAD+ salvage in LPS-activated, but not in unstimulated, macrophages. For instance, in unstimulated macrophages without cholesterol loading (white bars in the left panel of Figure 3E), the abundance of NAD+ did not change during NAMPT inhibition. However, when these cells were stimulated with LPS (white bars in the right panel of Figure 3E), NAD+ was reduced by NAMPT inhibition. In fact, this phenomenon was reported by Edward Pearce’s group (Cameron et al., Nat Immunol 2019). These authors found that NAD+ was reduced in LPS-stimulated macrophages and NAD+ levels were heavily dependent on the activity of NAMPT. Based on our data and published data from the Pearce lab, we feel that NAMPT blockade is sufficient to block NAD+ salvage in LPS-activated macrophages and NMNAT blockade is not necessary in this context.

Does PARP inhibition modulate the expression of ABCA1, in other words, is there a difference in cholesterol efflux from cells?

The reviewer raises an interesting point by inquiring if PARP inhibition may modulate the expression of ABCA1 and cholesterol efflux. Shrestha et al (JBC, 2016) found that inhibition of PARP1 in RAW264.7 macrophages and bone-marrow derived macrophages is linked to the augmentation of ABCA1 expression in a LXR-dependent manner. To determine if these findings can be reproduced in peritoneal macrophages, we inhibited PARP activity with rucaparib 1h prior to cholesterol loading and perform immunoblotting to measure ABCA1 expression. As shown in Supplementary Figure 2C, cholesterol loading of peritoneal macrophages significantly induced the expression of ABCA1. However, PARP1 inhibition did not modulate ABCA1 protein expression, which suggests that PARP1 inhibition in peritoneal macrophages may not play a significant role in modulating ABCA1 expression and cholesterol efflux.

The authors introduce HIF1 as a key pathway for the metabolic switch in macrophages, later shown NRF2 activation. HIF1 and NRF2 systems are interconnected. that should be discussed.

As suggested, we have modified the discussion. We discuss the involvement of NRF2 and HIF-1 in our model, as well as their interconnection.

The PAR blots should be shown as a whole with MW standards. These blots are either cut/cropped or are not showing the usual PARylation pattern. This a key point that needs to be cleared.

Thank you for pointing out the lack of MW standards in our PAR blots. We have now included these in all immunoblots in the revised manuscript. The PAR blots that we have presented are indeed cropped, specifically we have cropped all the bands below 63 kDa. Based on the titrated concentration of PARP1 inhibitor (AG-14361) shown in the uncropped original blot of Figure 2C from our initial submission (see below), we believed that all the bands below the 63 kDa marker are nonspecific as their expression persists at higher concentrations of AG-14361, whereas signals at higher molecular weight decreased.

As per PLOS ONE policy, authors are required to provide all original uncropped blots with MW standards as supplementary files. We have now provided the uncropped immunoblots with MW standards as supplementary files.

Please note that pharmacological PARP inhibitors block both PARP1 and PARP2, hence only by pharmacology it is not possible to distinguish between them. PARP2 is also associated with inflammation and cholesterol metabolism (PMID: 38698556). Please either introduce genetic studies or reshape the discussion. Furthermore, PARPs do modulate metabolism, hence the glycolytic switch may be related to PARP activation. This needs to be at least discussed.

Although the reviewer raises a valid point in pointing out pharmacological PARP inhibitors can block both PARP1 and PARP2, the lack of PARP inhibitor specificity was addressed in Figure 6 by siRNA knockdown of PARP1 in RAW 264.7 macrophages.

We would like to thank the reviewer for pointing out that PARPs can modulate metabolism and hence the glycolytic switch. As suggested, we have modified the discussion.

Typo:

line 215 „purines and pyrimidines derivatives” correctly: purine and pyrimidine derivatives.

We have fixed these typos.

line 335 Please add the missing reference.

We have added the missing reference.

---

## [Decision Letter · Decision Letter 1]

14 Jan 2025

Intracellular accumulation of free cholesterol in macrophages triggers a PARP1 response to DNA damage and PARP1 impairs lipopolysaccharide-induced inflammatory response

PONE-D-24-28324R1

Dear Dr. Cybulsky,

We’re pleased to inform you that your manuscript has been judged scientifically suitable for publication and will be formally accepted for publication once it meets all outstanding technical requirements.

Kind regards,

Zhiming Li, Ph.D.

Academic Editor

PLOS ONE

Additional Editor Comments (optional):

Reviewers' comments:

Reviewer's Responses to Questions

**Comments to the Author**

1. If the authors have adequately addressed your comments raised in a previous round of review and you feel that this manuscript is now acceptable for publication, you may indicate that here to bypass the “Comments to the Author” section, enter your conflict of interest statement in the “Confidential to Editor” section, and submit your "Accept" recommendation.

Reviewer #1: All comments have been addressed

2. Is the manuscript technically sound, and do the data support the conclusions?

Reviewer #1: Yes

3. Has the statistical analysis been performed appropriately and rigorously? 

Reviewer #1: Yes

4. Have the authors made all data underlying the findings in their manuscript fully available?

Reviewer #1: Yes

5. Is the manuscript presented in an intelligible fashion and written in standard English?

Reviewer #1: Yes

6. Review Comments to the Author

Reviewer #1: (No Response)

7. PLOS authors have the option to publish the peer review history of their article (what does this mean? ). If published, this will include your full peer review and any attached files.

**Do you want your identity to be public for this peer review?** For information about this choice, including consent withdrawal, please see our Privacy Policy .

Reviewer #1: No

---

## [Editor Report · Acceptance letter]

PONE-D-24-28324R1

PLOS ONE

Dear Dr. Cybulsky,

I'm pleased to inform you that your manuscript has been deemed suitable for publication in PLOS ONE. Congratulations! Your manuscript is now being handed over to our production team.

Kind regards,

on behalf of

Dr. Zhiming Li

Academic Editor

PLOS ONE